# Dealing with the Vicissitudes and Abject Consequences of Head and Neck Cancer: A Vital Role for Psycho-Oncology

Marco A. Mascarella [1,2], Gregoire B. Morand [1], Michael P. Hier [1,2,3,4], Alexander Mlynarek [1,2], Justine G. Albert [4], David Kissane [5,6] and Melissa Henry [3,4,7,*]

1   Department of Otolaryngology, Head and Neck Surgery, McGill University, Montreal, QC H4A 3J1, Canada
2   Department of Otolaryngology, Head and Neck Surgery, Jewish General Hospital,
    Montreal, QC H3T 1E2, Canada
3   Gerald Bronfman Department of Oncology, McGill University, Montreal, QC H4A 3T2, Canada
4   Lady-Davis Institute for Medical Research, Jewish General Hospital, Montreal, QC H3T 1E2, Canada
5   Cunningham Centre for Palliative Care Research, School of Medicine, University of Notre Dame Australia,
    Sydney, NSW 2010, Australia
6   Departments of Palliative Care, Cabrini Health and Monash Health, Monash University,
    Melbourne, VIC 3144, Australia
7   Department of Psychology, Jewish General Hospital, Montreal, QC H3T 1E2, Canada
*   Correspondence: melissa.henry@mcgill.ca; Tel.: +1-514-340-8222 (ext. 22252)

**Abstract:** Patients with head and neck cancer face important life-altering effects in appearance and function, affecting distress and quality of life and requiring the involvement of a multidisciplinary team. Psycho-oncology makes an important contribution to the field, as head and neck cancers carry a huge adaptational toll. To illustrate the value of this discipline, we report two cases of patients with advanced head and neck cancer for which the treatment-related body changes were of major significance. A commentary by the treating surgeons and psycho-oncologists precedes a general discussion about the clinical management of such patients. The article outlines strategies to address health literacy, doctor–patient communication, treatment decision-making, and emotional distress; placing the person at the center of oncological care. It calls for the broad application of principles of psychological first aid by healthcare professionals in oncology.

**Keywords:** body image; disfigurement; head and neck cancer; quality of life; shame; stigma; psycho-oncology

## 1. Introduction

Specialist multi-disciplinary care is essential in the management of patients with head and neck cancer, which afflicted 7400 Canadians in 2021, and resulted in 2100 deaths [1]. The morbidity resulting from these cancers can be profound, with striking impacts on quality of life through disfigurement, altered body image and sense of personhood. As a consequence, psycho-oncology makes an important contribution to such multi-disciplinary care provision, alongside key disciplines of surgery, medical and radiation oncology, speech therapy, dietetics, and physical and occupational therapy. To illustrate the value of this multidisciplinary approach, we report two cases of patients with advanced head and neck cancer for which the treatment-related body changes were of major significance. A commentary by the treating surgeons and psycho-oncologists precedes a general discussion about the clinical management of such patients. The article illustrates how important body image is to patients in head and neck oncology and how the viewpoint in the medical setting can be broadened when professionals in psycho-oncology are integrated into the treating team. It is the first article to our knowledge that presents clinical cases discussed first by surgeons and then by psycho-oncologists, illustrating the contribution of psycho-oncology to the understanding of patients and their management in medical clinics. It

also offers many concrete suggestions as to how the medical team can address issues of consent, communication, integration of information, health literacy, distress, and patient-centered care.

## 2. De-Identified Case Vignette 1

A 35-year-old single female, previously healthy, presented to her local hospital in a rural setting with a mass or fullness at the angle of the mandible on the left side. She was of Irish descent and presented a low socioeconomic status and low level of education with no cognitive impairment. Because the patient reported that this mass had persisted for more than one year when she was seen in our clinic she was scheduled for a biopsy of the lesion. When she saw the large team of physicians, residents and medical students enter the consultation room to see her, she fainted and dropped to the floor. The pathology was eventually not conclusive as not enough cells were present to make a reliable diagnosis. It was decided to perform an open biopsy in the office, which also provided an insufficient sample. A few days later, the patient presented again to the clinic with a severe deformity of her face, since the facial nerve on the side of the tumor had become paralyzed.

The patient was convinced that the mass was not a tumor, that surely some medication could get rid of it, and that her face would go back to moving like it used to. The medical team had multiple discussions with her to persuade her to have a repeat biopsy to confirm their suspicions of malignancy. She did not understand the medical terms presented to her and needed drawings and very simple explanations to help with her understanding. The patient initially refused a repeat biopsy but then presented to the emergency department because of severe facial pain. By then, several weeks had elapsed between the time of the initial presentation. With this new pain, she understood the severity of her situation.

After performing several scans and further discussion among the medical treatment team, surgical removal of this tumor of the parotid gland was deemed essential, with reconstruction using a flap from her arm in what would be a 10 h surgery. During the surgery, the facial nerve was seen to be encased by the tumor, so it had to be resected. This left the patient with severe disfigurement and facial paralysis. The final pathology revealed a malignant parotid tumor.

After the surgery, the patient underwent a 6-week course of radiation therapy. Unfortunately, the first post-treatment scan showed disease persistence, which could not be irradiated again, nor could it be resected. The tumor was sent for molecular analysis to identify potentially targetable mutations; but unfortunately, no mutations were identified, greatly limiting the potential efficacy of targeted therapeutics and/or immunotherapy. Confronted with this very hard reality, the patient sought palliative care and eventually succumbed to her disease.

*A Surgeon's Point of View*

One of the greatest challenges both prior to surgery and post-operatively was this patient's ability to understand her diagnosis, including the magnitude and severity of the disease and the potential for death from this cancer. Despite seeing multiple oncologists (surgical, medical and radiation), the priority for the patient focused on functional and cosmetic consequences (fear of complete facial paralysis, which is significant in itself) and not the extent of her disease. Even after surgery, when the surgical team delivered to her the news about the extent of her cancer and the need to escalate treatment with chemotherapy and radiation, her focus was on the recovery of her facial nerve rather than the risk of her own demise from the disease. All of this led to delays in procedures and treatment initiation, greatly limiting the prognosis of her condition initially thought to be favorable. A further issue was the patient's unwillingness to involve family and friends in her journey. The lack of direct psycho-social support from family and friends may have limited the medical team's ability to interact with and transfer important information to the patient.

## 3. De-Identified Case Vignette 2

A 78-year-old married Anglo-Saxon male patient presented initially with severe headaches over many weeks, for which CT imaging of the head was performed. The headaches prevented him from working in his accounting firm. This scan revealed the presence of a chondrosarcoma, a slowly growing, malignant tumor involving the base of the skull, usually managed by surgical excision. The patient was referred to a neurosurgeon, who performed endoscopic subtotal removal of the tumor.

The patient did well for a few months but unfortunately presented with a recurrence of this tumor. He underwent endoscopic sinus surgery with the removal of the recurrence. Post-operatively, the patient was recommended to undergo radiotherapy to the affected areas, to reduce the risk of re-recurrence. Unfortunately, about 1 year later, the patient presented again (and for the first time in our clinic) with a recurrence inside the nose, at the base of the nasal septum. A third resection was performed, resulting in an opening between the oral cavity and the nose. The patient had to be seen by dentistry to obtain a palatal prosthesis to be able to eat and speak properly.

About 9 months later, yet another recurrence was diagnosed, this time just below the skin of the nasal bridge. The patient underwent further surgery, with the removal of parts of his nasal bridge and reconstruction with a local flap, changing for the first time the appearance of his face. The surgery was initially successful with complete removal of the tumor seen at the time of surgery; however, a few months later, regrowth necessitated a complete removal of his nose (total rhinectomy).

After the removal of his nose, surgical options to reconstruct it were limited and esthetically challenging, so the patient was recommended to wear a tailor-made prosthesis based on a picture of his former nose.

This case report shows how a previously healthy 78-year-old patient presenting with headaches underwent a dramatic course of repeated operations and heavy medical treatment for a tumor that is generally slow growing, leading to severe and permanent changes to his physical appearance, speaking and eating abilities.

### 3.1. A Surgeon's Point of View

This patient's ability to cope with the severity and aggressiveness of necessary surgery to control and salvage him from dire consequences highlights some of the day-to-day challenges met in head and neck cancers. The trajectory from endoscopic, minimally invasive, initial surgery to total rhinectomy to salvage him from the effects of this chondrosarcoma represents a wide breadth of issues (e.g., disease-related, disfigurement, psycho-social). This is something that head and neck surgical oncologists deal with not infrequently. In some patients, when care is escalated, specifically surgery needed to perform organ resection, for instance, laryngectomy, subtotal glossectomy or total rhinectomy, patients can be divided into two groups: those that agree to the surgery and those that refuse, with often clear reasons for their choice (i.e., stigmatization, persistent suffering, dysphagia, aphonia). In this patient's case, and in contrast to the patient having undergone a parotidectomy above, the decision to actively pursue aggressive interventions was supported by both an understanding of his disease and strong psycho-social support from his family.

### 3.2. A Psycho-Oncologist's Point of View of These Two Cases

The face is central to a person's identity, sense of self, interpersonal communication, and relationship with others. Beauty can be alluring, lovely and facilitate relationships; in contrast, ugliness can be unsightly, hideous, and even repulsive to others. Head and neck cancers can so easily distort the face, causing disfigurement which embarrasses, social withdrawal and resultant isolation from others. More than facial appearance, head and neck cancers can also reduce the functionality of speech, sight, taste and smell, senses that are all important to personhood and identity. Thus, many losses may result from head and neck cancers, necessitating much grief and adaptation to a new level of function. At its worst, profound maladaptation can also occur when a sense of stigma develops and deep shame

about one's appearance causes psychiatric disorders such as depression, post-traumatic stress, anxiety, and states of demoralization, which at their worst can remove the desire to live. Ambivalence about treatment and non-adherence to recommended management can result.

In the first case vignette, facial nerve palsy becomes pre-occupying and intensely distressing to a young woman with parotid cancer encasing her facial nerve. We are told that she focuses on this facial appearance to the neglect of integrating information about the threat to her life that recurrence brings. Her cosmetic disfigurement could already be a major loss, potentially detracting from typical concerns at this age of intimacy and establishment of family life. While her surgical team has tried to cure or salvage her from disease progression, she may have sought time to deal with a sense of shame and the disease's impact on her identity. Her disease and its treatment may have wrought disastrous impacts upon her personhood. A tremendous amount of psychological support may be needed to help her adapt and make sense of her new reality.

The patient's hesitancy may be compounded by having undergone two inconclusive biopsies, which could have prompted some reluctance and mistrust in moving forward with a third and with the proposed treatments. The patient may have considered that most tumors of the parotid are benign, impacting her treatment decision-making process. The magnitude and severity of the patient's disease may have been experienced as unclear considering this. Weighing the pros and cons of surgery, she may have concluded that more time but with a poorer quality of life was undesirable.

This illustrates how important it is to consider the process of consent as part of a shared decision-making model, whereby consent moves from a passive process of information giving to one of bidirectionality where the patient actively participates in sharing their personal needs, values, and preferences [2]. Taking the time to listen to the patient's concerns and understanding their hesitancy or reluctance is key, leading to decreased decisional regret, increased adherence to recommendations, and decreased potential for litigation [3]. While clarifying any misconception and reassurance is paramount, one also needs to consider that the patient knows what is best within the context of his or her life. This includes whom he or she wants to involve in the medical consultations. As in the first clinical vignette, it is assumed that including family members or friends in the consultation room would have been facilitative. While social support has generally been implicated in better adherence to treatments, this is not always the case [4]. One may wonder about the quality of the patient's support system and their emotional and practical availability to be part of the consultations. One could also wonder if the nature of the medical encounter and its focus on the disease, or expectations thereof by the patient, may have played a role in the patient wanting to keep her personal life separate. One also needs to consider practical issues that weigh in the decision-making balance for patients, such as financial-, work-, and family-related impacts of the illness and treatments as well as barriers to care such as distance from the hospital and issues with transportation.

Psycho-social and palliative care providers are invaluable in assessing and addressing issues that may interfere with the patient's treatment decisions such as poor health literacy, lack of support system, practical issues, denial, and depression. In the end, after much ethical debate and mobilizing all potential viable psycho-social supports, even if difficult, the treating team must respect that the final decision lies within the patient and that palliative care can be chosen over other medical options. It may be necessary, in certain circumstances, to support the treating team in this process, as the patient's decision can oppose what they consider medically sound. This is especially important in the first clinical vignette, as in hindsight, the tumor may have been more aggressive and molecular testing eventually revealed no targetable mutations.

In the second case vignette, an older man must adapt to an obturator prosthesis and eventually a facial prosthesis to restore his cosmesis. What profound impacts on quality of life have occurred here! The cumulative losses that build from his slowly yet steadily progressing tumor, impacting upon his speech, mastication, eating and swallowing, before

finally his nose is sacrificed in this never-ending attack on his personhood. Interestingly, so intense has the focus been on managing his disease that we are not told whether he is married, a parent, or who is supporting him on such a demanding journey. Instead, we are only told very generally that his family was instrumental in supporting adherence to treatment recommendations, which is also restricted to the medical domain rather than allowing for an enriched description of this man, the significance of his family relationships, and how the decision-making was made as a family unit. There can be no doubt that the role of the psycho-oncologist is central to the support and adaptation of these two patients whose lives were devastatingly impacted by these ravaging cancers. Integrated care would also mean for the surgeons to adopt a more person-centered rather than disease-focused stance in their clinical practice, distinguishing themselves from a purely surgical model to espouse an interdisciplinary-based care approach. This means integration of empathy and concern into patient encounters and broadening their view to include the person in his or her context, which can be healing in and of itself.

The two cases illustrate the unique needs of patients at different stages in their lives and the significant impact of developmental factors on how these patients are approached and how they process their cancer diagnosis. The first case illustrates the unique concerns of an adolescent and young adult (AYA) population, with potential issues around identity, autonomy, body image, sexuality, and interpersonal relationships that are central to adolescents and young adults [5]. Survivorship can be particularly problematic, as the illness can interrupt important developmental milestones and be isolative vis-à-vis normal peer groups and the healthcare system. This is compounded by the increased uncertainty that relates to the effectiveness and availability of treatments as well as the lack of evidence-based prognostic information in AYAs [6]. AYAs face increased financial toxicity, as well as competing decision-making around employment opportunities, educational attainment, and romantic and family pursuits. This can be exacerbated by the knowledge of long-term treatment sequelae such as secondary malignancies, cardiovascular disease, endocrine dysfunction, neurocognitive deficits, fertility issues, sexual dysfunction, and a decline in physical fitness. The treating team needs to consider these factors in their approach to AYA patients with cancer, their discussions around treatment options, and survivorship care [6].

Health literacy and high levels of distress may impact the understanding that patients achieve about cancer and its consequences and may have played a particular part in the first case vignette. Health literacy is defined as "the capacity to obtain, process, and understand basic health information and services needed to make appropriate health decisions" [7]. Health literacy is viewed to incorporate three hierarchical dimensions: functional (reading written instructions and completing health-related forms), interactive (ability to act on information received through communication), and critical (advanced cognitive and social skills for critically analyzing and applying health information) [8,9]. Past research suggests that lower health literacy is associated with a lack of screening and preventative behaviors, a longer time elapsing between symptom identification and seeking medical attention, difficulties understanding and processing cancer-related information, lower levels of information-seeking behaviors, impairments in risk perception, non-compliance, lack of patient–provider communication, unfavorable experience with care, and a lower quality of life [10,11]. Patients with lower health literacy tend to present with lower educational attainment, lower income, and live in a rural area [11], as described in the clinical vignette.

While there exist screening tools for health literacy [12,13], some have criticized their use in medical settings [14], proposing instead universal health literacy precautions [15] guided by strong communication skills [16,17] and a shared decision-making model placing patients at the center of care [17,18]. A recent literature review has identified five aspects of patient-centeredness: espousing a biopsychosocial perspective; considering the 'patient-as-person'; sharing power and responsibility; creating a therapeutic alliance; and considering the 'doctor-as-person' [19]. The World Health Organization has conceptualized people-centered care as placing the person within their larger context of life, including the family, community, and country settings [20]. Evidence-based tools to facilitate communication can

include simple, jargon-free communication, the use of analogies or metaphors to help with understanding and retention, the teach-back method, patient decision aids, consultation audio recordings, and question prompt lists or sheets [11]. Unfortunately, none of these tools were used by the surgeons in the presented clinical cases.

The teach-back method is a series of iterative steps used to ensure that the provided medical information is understood correctly [21]. It invites the patient to summarize back to the clinician what they have understood about their diagnosis and the management plan. This will confirm their ability to integrate the information presented to them and begin to reveal their emotional reaction to this news. The physician can then clarify and re-explain if needed, asking again to explain what was understood with the process repeated until the information is recalled correctly. Can a family member attend with them, take notes and support them during these consultations? Might the clinician audio-record the consultation so that critical information can be listened to again [22]? The distressed patient may need encouragement to play back such a recording two or three times before they begin to process what is recommended by the clinician [23,24]. Decision aids can present information visually and incorporate value clarification exercises to help clarify values, treatment outcome preferences, and risk/benefit trade-offs that the person is willing to make to minimize decisional regret [18]. Question prompt lists or sheets are either a structured fixed or selected set of items, questions, and concerns that patients can use to prepare for their oncology consultations [25].

The setting and way in which information is transmitted by the medical team, coordination of care, and transitions in care, can all contribute to the stress of a cancer diagnosis and treatments. As we could see from the two clinical vignettes, the medical encounter was more disease- rather than person-centered, encouraging a top-down rather than a shared approach to decision making. In the first case vignette, the patient fainted when she saw her team of doctors for the first time, perhaps out of fear of the potentially life-threatening nature of her condition or because she was intimidated by the large team coming into the room to see her. This additional contextual stress can affect executive functions, which are particularly important in the process of assimilating medical information and making treatment decisions. One may want to minimize any additional stress experienced by patients, such as having one physician walk into the room, introduce themselves and the context, and then ask to introduce the other members of the team; so as to establish a sense of safety and control in this already stressful setting. Strong emotions stimulate amygdala- and ventral striatum-centered circuits in the brain, phylogenetically involved in promoting survival of the species and an immediate fight-or-flight response over slower more deliberate reasoning. The frontoparietal network, more specifically the dorsolateral prefrontal cortex (dlPFC), has evolved through our ontological history and is involved in reasoning and higher-order cognition (e.g., selective attention, working memory, cognitive control), as well as top-down regulatory control of emotions and behaviors. The dlPFC helps maintain attention during distracting emotional stimuli. dlPFC functions are related to mood and anxiety disorders and may be driven by a combination of genetic predisposition and early childhood experiences [26]. Considering this knowledge of brain circuitry and in line with the Hippocratic Oath, one cannot overestimate how central the caring context and relationships are to both healing and optimizing decision-making in the face of cancer. An increased focus on person-centered care by the health care professionals in the context of these two clinical vignettes, and more particularly the first one, may have gone a long way in alleviating the patients' distress and better aligning therapeutic objectives, perhaps contributing to heightened trust in the relationship. Proper psycho-social referrals could have followed as stress and avoidance were high.

One would want front-line health care workers in oncology to be trained in and apply principles of psychological first aid [27] considering the potentially traumatic context of cancer, providing safety, stability, and resources to patients in this very stressful time, especially when treatment options threaten vital functions physically and socially such as in head and neck cancers. Staff may need to be trained to help address basic needs such as

food and shelter, listen to patients' stories with empathy, foster hope and connection, and give basic suggestions for coping, all of which can affect stress responses and treatment decision-making. Part of the front-line medical team's, including the surgeon's, role, in this case, may be to validate the experience of young adults presented with disfiguring surgery, express hope in their ability with time to adapt and reconstruct a new life post-surgery, present the possibility of meeting other patients that have gone through and readjusted to this type of surgery, present (personal, hospital, and community-based) venues for support if needed in the process of readjustment, and affirm patients' identity and aspirations to lead a self-fulfilled life. One needs to develop a perspective beyond the hospital setting, including the person in his/her natural and historical context.

## 4. General Discussion

Many patients are challenged greatly by the development of head and neck cancer and its impact on their quality of life [28]. They can struggle to cope and readily develop anxiety, demoralization, depression, shame, stigma, and resultant avoidant behaviors [29]. Body image disturbance is especially problematic [30,31]. Intimacy and sexual functioning can then become impaired [32]. When embarrassment and humiliation result in a feeling of shame, the sense of disgrace and dishonor that the patient feels may create an urge to cover oneself, hide or avoid any public scrutiny, well exemplified by the *Phantom of the Opera*. When guilt and regret exist over behaviors that have contributed to the development of cancer, such as the use of tobacco and alcohol, social disapproval can be felt and impact a patient's openness to treatment. Disfigurement and spoiled body image contribute strongly to shame and stigma among patients with head and neck cancer [33].

If we are to advance the care of these patients with head and neck cancer, routine measurement of these aspects of their quality of life may prove helpful followed by prompt referrals to psycho-oncological supports. Fortunately, reliable and well-validated measures have been developed to assist in such measurement of their body image [34] and any sense of shame or stigma [35] resulting from their head and neck cancer or its treatment. Measures of body image, depression and anxiety can be included in distress screening, considered an essential component of patient-centered care, and endorsed as the sixth vital sign [36,37]. These measures can identify patients early on with high levels of distress, prompting the use of a collaborative care model including pharmacotherapy and psycho-oncology. In patients using avoidance or denial, an anti-anxiety agent combined with psychotherapy (specifically CBT, graded exposure with relaxation, and existential psychotherapy) can help decrease avoidance-based coping and increase approach-based coping necessary to face the adaptational stress of treatments for head and neck cancer. Exploring underlying issues related to hesitation or refusal of treatments is a necessity to consent, as seen more particularly in the first case vignette and may in some cases require a referral to psycho-oncology. Recognition of these issues empowers appropriate engagement of support services to optimize coping, sense of support and overall adaptation in patients. Peer support can be particularly valuable in the process of decision-making and rehabilitation, allowing patients to connect to peer mentors who have faced similar challenges and can promote resilience and hope. Patient narratives were also integrated into decision aids with positive effects, especially in patients with lower literacy [38]. Otolaryngology clinics need to work on enhancing access to psycho-oncological care and timely referral, through better integration of psycho-oncology services in medical clinics, mental health stigma reduction campaigns, promotion of psycho-oncological services, communication skills training of health care professionals, and continuing education of the medical team on distress management [39], and adoption of a person-centered approach.

Acceptance and commitment therapy (ACT) is a third-wave cognitive behavioral therapy, which may have particular utility in helping patients with head and neck cancer. Attention to grief and emotional responses, cognitive expectancies about the diagnosis and treatment, specific illness representations that include body image and any sense of disfigurement, confidence about adjusting to these losses and changes, and access to

support and assistance over time become crucial dimensions of therapeutic models [40]. Other third-wave approaches include Mindfulness-Based Cognitive Therapy (MBCT) and Mindfulness-Based Stress Reduction [41]. Trials of ACT are currently proceeding internationally to examine its efficacy in helping cohorts of patients with head and neck cancer. Its emphasis on acceptance rather than allowing avoidant behaviors to develop makes it particularly pertinent to the issues that predominate in head and neck cancers.

Our clinical vignettes have exemplified the ambivalent and avoidant behaviors long known to occur in patients with head and neck cancer—the profound distortion of a young woman's face from facial nerve palsy associated with a parotid cancer, and the opening up of the palate from maxillary surgery, with later loss of the nose as the chondrosarcoma steadily destroys the man's face. These extreme examples highlight the extraordinary challenges that patients can face in the setting of head and neck cancer. Such patients require sustained support over time, often mixed psychotropic and psychotherapeutic palliation, and this includes help as they negotiate treatment decisions as well as beyond the initial treatment into social re-integration. Maintaining the dignity of the person and their sense of the continued meaning and value of their life is crucial, especially when their disease progresses, and they must eventually face their death.

Outcomes for these patients may not have altered with greater psycho-oncology involvement. We need to be modest in the face of aggressive disease. Nevertheless, whether a patient is adherent or non-compliant with recommended management, psycho-social support and treatment of distress is crucial as part of the multi-disciplinary approach. Early involvement of psychology/psychiatry/social work and a person-centered approach ought to be a sine qua non-integral component in the management of such complex clinical cases, as when we delve into people's experiences there are often many more layers to discover and address.

**Author Contributions:** Conceptualization, M.A.M., G.B.M., M.P.H., A.M. and M.H.; Methodology, M.A.M., G.B.M., M.P.H., A.M. and M.H.; Writing—original draft preparation, M.A.M., G.B.M., M.P.H., A.M., J.G.A., D.K. and M.H.; Writing—review and editing, M.A.M., G.B.M., M.P.H., A.M., J.G.A., D.K. and M.H. All authors have read and agreed to the published version of the manuscript.

**Funding:** There was no external funding related to this paper. MH is receiving a Fonds de recherche santé—Québec Senior Salary Award for her research.

**Institutional Review Board Statement:** Ethical review and approval was not needed for this paper, as the cases have been collated from different sources and all identifying information has been changed.

**Informed Consent Statement:** Informed consent was not needed for this paper, as the cases have been collated from different sources and all identifying information has been changed.

**Data Availability Statement:** The data presented in this study are available on request from the corresponding author.

**Conflicts of Interest:** The authors have no conflict of interest to declare.

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
