# Peer review of "Dealing with the Vicissitudes and Abject Consequences of Head and Neck Cancer: A Vital Role for Psycho-Oncology"

_curroncol, doi:10.3390/curroncol29090527_

Round 1

Reviewer 1 Report (Previous Reviewer 1)

Dear Authors,

Thank you for the submission. This was an interesting paper to read. The topic has high clinical relevance. I enjoyed reading this manuscript and would be happy to accept in present form.

Author Response

Reviewer 2 Report (Previous Reviewer 3)

Unfortunately the revisions to this manuscript do not significantly improve the overall content, clarity, merit, or overall usefulness to the field.

Specifically, the authors still do not link the first vignette to poor health literacy, do not give a diagnosis, prognosis, or discuss palliative care as a reasonable option. In one section, they seem to have copied and pasted reviewer comments into the manuscript.

This manuscript could be made significantly stronger if the vignettes aligned with the topics presented (health literacy, distress, shared-decision making, teach back, etc.) and informed readers of how these issues present in patient encounters, how providers identify and assess and ultimately treat these issues. The current manuscript lacks this continuity and usefulness to providers.

Author Response

This manuscript is a resubmission of an earlier submission. The following is a list of the peer review reports and author responses from that submission.

Round 1

Reviewer 1 Report

Dear author,
The article cannot be published in this form as it is not of significant clinical or surgical interest and does not add anything more to what is known about the importance of multidisciplinary management of the patient with head and neck cancer.

Reviewer 2 Report

This manuscript is well written and presents an important case study highlighting the psychosocial challenges that many head and neck cancer patients face, and the need for psychosocial care throughout the cancer trajectory. This is an important issue that often goes undiscussed both in the literature and in care settings. Below are a few suggestions to improve the manuscript. 

  1. I think some discussion regarding the unique needs of AYA (adolescents and young adults) with cancer must be addressed. The fact that these patients are in 2 very different life stages and the significant impact of developmental factors in how these patients are approached and how they process their cancer diagnosis must be acknowledged. AYA have distinct needs that differ from older adults, but this consistently fails to be acknowledged in health care settings. The health care team needs to be aware of and tailor their approach to communication in consideration of these developmental factors. AYA face additional challenges such as financial toxicity, difficulties with employment, educational attainment, etc. which older adults with cancer typically do not need to consider. This is a crucial factor to consider in how the patient is approached and their priorities for care. 
  2. Lines 231 - 234: Measures exist but how often do patients actually get referred to a psychologist or other supportive care expert? It is not uncommon for patients/survivors to express frustration in their health care team failing to take action (e.g. refer to a psychologist) despite high scores/levels of distress. How could this be addressed by the health care team?
  3. Lines 237 - 239: What about peer support? The crucial role of patient mentors is consistently underestimated by health care providers. This is particularly true for AYA patients who report opportunities to connect with peers who've faced a similar experience to be a high priority and often an unmet need.
  4. Line 268: "sine qua non" - would be helpful to clarify what this means.

Reviewer 3 Report

Dealing with the vicissitudes and abject consequences of head and neck cancer: A vital role for psycho-oncology 

This manuscript seeks to demonstrate the importance of psychosocial oncologists, specifically in head and neck cancer. Two case vignettes are presented and an extended discussion of the cases are given. 

There are multiple concerns with this manuscript, specifically with vignette 1. Authors note that the patient had undergone 2 inconclusive biopsies, then note that she was reluctant to have a third biopsy. Seemingly, authors relate this to poor health literacy, when in fact it may be related to loss of trust in the medical system, denial, depression, or appropriate hesitancy given that 2 biopsies had already been performed. Additionally, most tumors of the parotid are benign, which could also impact the patient’s decision-making process. Clarification that this is truly due to poor health literacy needs demonstrated. 

Section 2.1 notes the patient did not understand the “magnitude and severity of the disease” and without a defined diagnosis, it is unclear to me as well. What was her diagnosis and prognosis? What would her expected quality of life look like had she immediately followed all treatment recommendations? Would treatment give her more time with poorer quality of life? 

If the potential for death is high as stated, is it not justifiable to not want to pursue treatment? Or focus on Palliative care instead? Especially given the molecular analysis did not identify any targetable mutations. Is no treatment a viable option in this case?

Medical providers are trained to fix, cure, and treat conditions. However, patients are the ones who must live with the consequences (scars, disfigurement, functional impairment, etc.) of the disease and its treatment. It seems imperative to listen to their concerns and fully understand their hesitancy or reluctance. Sometimes the physician “knows” what is right for the patient, however, we have to trust that the patient knows what is best for them and give them autonomy over the decision-making process (unless of course there are concerns for competency and capacity). Psychosocial and Palliative Care providers should be able to assess and address issues of poor health literacy, lack of support system, aspects of denial, depression, etc. that may get in the way of a patient making treatment decisions. However, none of that was elaborated on in relation to this vignette or the extended discussion of this patient.  

This is further exemplified by the comments about her “unwillingness to include family and friends in her journey.” Do we have more context as to the relationships with these family and friends? Does she actually have support but chooses not to include them or does she not have a strong support system? Is the support system: actually supportive? Abusive? Are they close by or live far away? Working 2 jobs with less availability? It would be helpful to further clarify what is meant by that statement. 

In many ways, this vignette reads as a very paternalistic approach to medicine where the physicians are right and a patient who is not compliant needs to be told what to do rather than engaging in joint decision making or a further clarification as to why the patient is reluctant. 

In section 3.2 in the discussion of vignette 1 it is stated, “Yet her cosmetic disfigurement is already a major loss, potentially detracting from her hopes for romance, marriage and a conventional family life” and goes on to mention her “femininity and womanhood” please clarify that these are in fact her concerns and include in vignette as reasons for her reluctance, otherwise it reads as gendered and sexist. If these are not her concerns, they could be included as normative developmental challenges (intimacy versus isolation) and discuss how it would not be uncommon for someone this age to have concerns about these issues as it is developmentally appropriate at this age. Hence the focus on age and not her sex/gender identity. 

For the older male patient, we are told that there has been so much intense focusing on his disease management that providers did not know if he was married, was a parent, or who he was supported by. In the young female patient vignette, we are told that she was “unwilling” to include her support system. Could the older male patient also be unwilling? Or could the younger female patient also have been focused on things other than letting providers know if she were married, a parent, or who she was supported by? In the young female patient, being “unwilling” to have support was viewed as an impediment to treatment. Why is it not viewed as an impediment to both? Only because the older male patient followed the treatment recommendations? 

Authors should include educational levels of the 2 patients and associated risk factors (education level, socioeconomic status, disability, race/ethnicity, cognitive impairment, language or cultural barriers, etc.) for each if they are trying to demonstrate poor health literacy. 

The sections on health literacy, teach back, and therapeutic approaches to psychological difficulties were all very strong. Were any of these approaches used with the patients in the 2 vignettes? If so, please elaborate regarding how it went. If not, elaboration about speculated treatment approaches could be helpful.  

There is no doubt in my mind that psychosocial care is imperative in working with patients with cancer. Head and neck cancers present their own significant challenges to be addressed. Psychosocial providers are able to delve into the barriers and challenges that patients may face (body image changes, functional impairment, symptom management, communication with medical teams, addressing psychological challenges such as distress, anxiety, depression, etc.). The discussion about psychological treatments is strong, however, it does not seem connected to the vignettes presented. Additionally, as noted above, the vignettes are problematic and should undergo substantial revision to make them fit with the discussion presented or include different vignettes where authors can demonstrate the psychosocial approaches being used.